# Efficacy and Safety of Ablative Therapy in the Treatment of Patients with Metastatic Pheochromocytoma and Paraganglioma

**DOI:** 10.3390/cancers11020195

**Published:** 2019-02-07

**Authors:** Jacob Kohlenberg, Brian Welch, Oksana Hamidi, Matthew Callstrom, Jonathan Morris, Juraj Sprung, Irina Bancos, William Young

**Affiliations:** 1Division of Endocrinology, Diabetes, Metabolism, and Nutrition, Mayo Clinic, 200 First Street SW, Rochester, MN 55905, USA; Kohlenberg.Jacob@mayo.edu (J.K.); Wyoung@mayo.edu (W.Y.J.); 2Department of Radiology, Mayo Clinic, 200 First Street SW, Rochester, MN 55905, USA; Welch.Brian@mayo.edu (B.W.); Callstrom.Matthew@mayo.edu (M.C.); Morris.Jonathan@mayo.edu (J.M.); 3Division of Endocrinology and Metabolism, University of Texas Southwestern Medical Center, 5323 Harry Hines Blvd., Dallas, TX 75390, USA; Oksana.Hamidi@utsouthwestern.edu; 4Department of Anesthesiology, Mayo Clinic, 200 First Street SW, Rochester, MN 55905, USA; Sprung.Juraj@mayo.edu

**Keywords:** radiofrequency ablation, cryoablation, percutaneous ethanol injection, neuroendocrine tumor, minimally invasive procedure, percutaneous ablation

## Abstract

Metastatic pheochromocytoma and paraganglioma (PPGL) are incurable neuroendocrine tumors. The goals of treatment include palliating symptoms and reducing tumor burden. Little is known about the use of radiofrequency ablation (RFA), cryoablation (CRYO), and percutaneous ethanol injection (PEI) to treat metastatic PPGL. We performed a retrospective study of patients age 17 years and older with metastatic PPGL who were treated with ablative therapy at Mayo Clinic, USA, between June 14, 1999 and November 14, 2017. Our outcomes measures were radiographic response, procedure-related complications, and symptomatic improvement. Thirty-one patients with metastatic PPGL had 123 lesions treated during 42 RFA, 23 CRYO, and 4 PEI procedures. The median duration of follow-up was 60 months (range, 0–163 months) for non-deceased patients. Radiographic local control was achieved in 69/80 (86%) lesions. Improvement in metastasis-related pain or symptoms of catecholamine excess was achieved in 12/13 (92%) procedures. Thirty-three (67%) procedures had no known complications. Clavien-Dindo Grade I, II, IV, and V complications occurred after 7 (14%), 7 (14%), 1 (2%), and 1 (2%) of the procedures, respectively. In patients with metastatic PPGL, ablative therapy can effectively achieve local control and palliate symptoms.

## 1. Introduction

Pheochromocytoma (PHEO) and paraganglioma (PGL) are rare neuroendocrine tumors that arise from the adrenal medulla and autonomic paraganglia, respectively. The incidence in the United States is 500 to 1600 cases per year [1]. While the majority of PHEO and PGL (PPGL) do not metastasize, 2–13% of PHEO and 2.4–50% of PGL are metastatic [2,3,4]. PPGL are considered metastatic when nodal or distant metastases are identified [5].

Patients with metastatic PPGL most frequently present with manifestations of catecholamine excess [6]. However, patients also present with symptoms from tumor-related mass effect or incidentally following imaging performed for an unrelated indication [6]. The natural history of metastatic PPGL is highly variable: for some patients it progresses rapidly, but others have prolonged survival [6]. Patients with metastatic PPGL have a median overall survival of 24.6 years and a 5-year mortality rate of 37% [6,7]. There are no differences in disease-specific mortality between metastatic PHEO and PGL [6].

A number of localized and systemic therapies are currently used to treat patients with metastatic PPGL; however, none of the options is curative. The goals of treatment are to reduce manifestations of catecholamine excess, palliate metastasis-related pain, and improve prognosis by treating lesions likely to progress and become symptomatic.

Given the rarity of metastatic PPGL, many of its treatment options have not been extensively studied. In particular, little is known regarding the use of thermal and chemical ablation to treat metastatic PPGL. Therefore, the objectives of this study were to investigate the efficacy and safety of radiofrequency ablation (RFA), cryoablation (CRYO), and percutaneous ethanol injection (PEI) in the treatment of patients with metastatic PPGL. In this study, we found that ablative therapy can be successfully used in patients with metastatic PPGL to achieve radiographic local control, palliate metastasis-related pain, and reduce symptoms of catecholamine excess. We also found that the majority of patients treated with ablation did not experience any procedure-related adverse effects.

## 2. Results

### 2.1. Patient Demographics and Clinical Presentation

All ablations were performed between June 14, 1999 and November 14, 2017. Thirty-one patients (*n* = 22, 71% women) with metastatic PPGL (*n* = 24, 77% with PGL, and *n* = 7, 23% with PHEO) underwent treatment of metastatic lesions with RFA, CRYO, or PEI (Table 1).

### 2.2. Ablation Sessions

Thirty-one patients underwent a total of 69 ablation sessions to treat 123 metastatic lesions. Of the 123 metastatic lesions, 114 were treated with percutaneous ablation and 9 were ablated intra-operatively. A total of 42 RFA, 23 CRYO, and 4 PEI were performed. Seven patients underwent more than one type of ablation during the same session (e.g., RFA of one lesion immediately followed by CRYO of a separate lesion) for a total of 57 procedural sessions (Table 2).

### 2.3. Ablation Session Indications

Manifestations of catecholamine excess were present prior to 18 (37.5%) procedural sessions and absent prior to 30 (62.5%) procedural sessions (Figure 1). Eleven (22%) procedural sessions were performed to treat 17 painful lesions (Figure 2). Five of the ablated lesions were in high-risk anatomic locations, for example an osseous lesion in close proximity to the spinal cord. In general, ablation was performed to achieve local oncologic control and to mitigate the risks associated with local tumor progression.

### 2.4. Pre-Ablation Adrenergic Blockade

Functioning metastases were targeted in 31 (63%) of the 49 procedural sessions that had a known pre-ablation biochemical status. Of these, 5 (16%) had an adrenergic biochemical phenotype, 25 (81%) had a noradrenergic biochemical phenotype, and 1 (3%) was dopaminergic only.

Of the 31 procedural sessions performed to treat functioning metastases, α-adrenergic blockade was given prior to 29 (97%), β-adrenergic blockade was given prior to 27 (87%), and the tyrosine hydroxylase inhibitor—metyrosine—was given prior to 23 (74%). Of the 26 procedural sessions performed to treat metastases with a non-functioning or unknown hormonal status, α-adrenergic blockade was given prior to 10 (50%), β-adrenergic blockade was given prior to 4 (20%), and metyrosine was given prior to 1 (5%). The purpose of pre-ablation adrenergic blockade was to reduce hemodynamic variability due to catecholamine release during the procedure.

We initiated α-adrenergic blockade generally 7–14 days pre-ablation, with phenoxybenzamine the most frequent medication prescribed. For patients with heart rates persistently greater than 80 beats/minute, we administered β-adrenergic blockade 2–5 days before ablation. Adrenergic blockade was titrated to effect and if patients remained hypertensive, a calcium channel blocker was added. Metyrosine therapy was used in patients with anticipated significant catecholamine release who were not controlled with α-adrenergic blockade. The initiation of metyrosine was based on clinical judgment and was often prescribed for patients undergoing ablation of multiple lesions during the same procedural session, patients with significantly elevated catecholamines and metanephrines, and patients who had significant hemodynamic variability during previous ablation sessions. Metyrosine was administered orally in the form of 250 milligram (mg) capsules. The Mayo Clinic protocol for short-term metyrosine preparation for patients already taking α-adrenergic blockade is to start metyrosine 4 days pre-ablation at a dose of 250 mg 4 times daily (QID) and up-titrate as tolerated to 500 mg QID 3 days pre-ablation, 750 mg QID 2 days pre-ablation, and 1000 mg QID the day prior to ablation, with the last dose (1000 mg) given on the morning of the ablation.

### 2.5. Ablated Lesions

A total of 123 lesions were ablated, with a median of 3 ablated lesions per patient (range, 1–15). Of the 123, 63 (51%) were osseous and 54 (44%) were hepatic. Six (5%) were abdominal/pelvic non-hepatic lesions including 4 retroperitoneal lesions, 1 abdominal lymph node, and 1 peri-urethral soft tissue metastasis (Table 2).

### 2.6. Technical Success

Technical success was achieved in 94 (94%) of the ablated lesions. For 19 lesions, the status of technical success was unknown due to lack of post-ablation imaging. Four lesions were not assessed for technical success because the patient died during the ablation (Table 2).

### 2.7. Radiographic Outcomes

Of the 123 ablated lesions, 80 (65%) were included in radiographic outcomes analysis. Of the 43 excluded lesions, 37 lacked post-procedural imaging (many patients were referred solely for therapy and received post-ablation care elsewhere) and technical success was not achieved for 6. Overall, local control was accomplished in 69 (86%) of the ablated lesions (Table 3).

Fourteen (45%) of the 31 patients were treated with systemic therapy, including cytotoxic chemotherapy, molecularly targeted therapy, and radiolabeled somatostatin analogs. Several of these patients had ablations performed while receiving systemic therapy (Table 1). Overall, local control was achieved in 36 (78%) lesions that were ablated in patients who had received systemic therapy at any time during the disease course.

For lesions with local control, the median total duration of follow-up was 26 months (range, 2–163 months) after ablation. With the exception of 1 peri-urethral soft tissue lesion, all metastases treated with CRYO were osseous lesions.

Of 80 ablated lesions, 11 (14%) progressed at a median of 16 months (range, 6–69 months) (Table 3). All 11 lesions that progressed occurred in only 5 patients. Of the patients who underwent technically successful hepatic ablations, additional hepatic metastases developed outside the area of ablation in 10 (67%) instances at a median of 16 months (range, 2–24 months) following ablation.

### 2.8. Symptom Outcomes

Technical success was achieved for 16 of the 17 painful metastases that were ablated. Symptom response to ablation was documented for 8 (50%) of the 16 lesions with technical success. Of those 8, all (100%) had improvement in pain following ablation (Figure 2).

Of the 18 procedural sessions performed to treat manifestations of catecholamine excess, 13 were excluded from symptom outcome analysis due to either lack of technical success, simultaneous resection of the primary tumor, intra-ablation death, or unknown post-procedural symptoms. Of the remaining 5 procedural sessions, symptoms of catecholamine excess improved following 4 (80%) (Figure 1 and Figure 3).

### 2.9. Ablation Session Complications

No complications occurred following 33 (67%) procedural sessions. Clavien–Dindo Grade I minor complications occurred following 7 (14%) procedural sessions and included transient fever, persistent ablation site pain, and minor bleeding not requiring intervention. Grade II complications developed following 7 (14%) procedural sessions and the most common intervention for these patients was intravenous blood pressure medications. Overall, overnight continuous hemodynamic monitoring for labile hemodynamics was required following 6 procedural sessions. One patient had a Grade IV complication following RFA of four metastatic lesions within the retroperitoneum. He initially did well post-ablation but approximately one week later developed gastrointestinal bleeding and required surgery although the sites of gastrointestinal bleeding were not suspected to be directly related to his ablation procedure. One patient had a Grade V complication and died from a likely argon gas embolism during CRYO of 4 osseous lesions (Table 4).

### 2.10. Long-Term Outcomes

Of the 31 patients, 10 (32%) died secondary to metastatic PPGL (median age at death, 53.5 years; range, 31–75 years) and 21 (68%) were alive at the time of last follow-up (median 60 months; range, 0–163 months) (Table 4). Of the 14 patients treated with systemic therapy, 7 (50%) died secondary to metastatic PPGL.

## 3. Discussion

In this study, we described the indications, efficacy, and safety of thermal and chemical ablation in the treatment of patients with metastatic PPGL. We found that the main indications for ablation therapy were to palliate metastasis-related pain, reduce manifestations of catecholamine excess, stabilize metastases in high-risk anatomic locations, and achieve local oncologic control to prevent risks associated with local tumor progression. We found that 94% of ablations were technically successful, and radiographic local control was achieved following 86% of those ablations. Ablative therapy was also successful in reducing metastasis-related pain and treating manifestations of catecholamine excess. Overall, ablative therapy was safe for the majority of patients. However, complications were noted after one third of procedures. Most of these complications were minor with no long-term sequelae. However, there was 1 ablation-related death.

In our study, the most common indication for ablation therapy was to achieve local control. Overall, radiographic outcomes were similar for type of ablation (CRYO versus RFA) and location of metastases (osseous versus hepatic). Of the lesions treated with thermal ablation, osseous metastases were treated with both RFA and CRYO, while hepatic metastases were treated only with RFA. We found that for metastases treated with RFA, radiographic local control was achieved more frequently for hepatic lesions (94%) than osseous lesions (74%). In contrast to hepatic lesions, osseous lesions can be more challenging to ablate because of their location and the propagation of thermal energy in the bony matrix. Further, it is easier to visualize and confirm complete treatment of hepatic metastases than osseous metastases because of the inherent differences between the two environments.

Local control was achieved in 78% of ablated lesions in patients who received systemic therapy at any time during the disease course. In contrast, local control was achieved in 97% of ablated lesions in patients who never received systemic therapy. This suggests that patients requiring systemic therapy have more aggressive disease that is more likely to ultimately progress. However, ablative therapy can still be an effective means to achieve local control in lesions in high risk anatomic locations in patients who require systemic therapy.

The previously published literature on ablative therapy for the treatment of metastatic PPGL is limited to single patient reports and small case series, which makes it challenging to compare our findings [8,9,10,11]. However, there is suggestion that interventional radiology techniques may delay the development of serious skeletal-related events in patients with metastatic PPGL [12].

Regarding radiographic response to ablation, prior studies used different definitions of local control than our study. For example, in a case series of 6 patients with metastatic PHEO treated with RFA, complete ablation was achieved in six of the seven ablated metastatic lesions at a mean follow-up of 12.3 months [10]. In this report, complete ablation was defined as a lack of enhancement within the ablation zone on follow-up CT [10]. Moreover, in a previous report from our institution, local control was defined as the absence of metastases in the treated region and was observed in 15 of 27 (56%) ablated lesions.

However, in our current study of 31 patients who had 123 lesions ablated, we defined local control as either no evidence of disease in the area of ablation or decreased tumor burden in the area of ablation when compared to pre-ablation imaging. Our definition of local control was chosen for several reasons—most significantly to provide a clinically meaningful classification. PPGL are known to have a slow response to therapy: following successful treatment, a residual mass is often present on imaging [13]. However, this alone does not represent treatment failure because imaging findings that indicate local control of PPGL following treatment include decreased tumor enhancement and decreased tumor size [13]. Additionally, since metastatic PPGL is incurable, the goal of therapy is to reduce tumor burden to palliate symptoms and intervene early when a patient has lesions expected to progress and become symptomatic.

Only 14% of the 80 technically successful ablated lesions had radiographic progression at a median of 16 months (range, 6–69 months) after ablation. Due to the limited number of technically successful ablations with subsequent radiographic progression, we were not able to identify predictors of lesion progression following ablation.

Our results suggest that ablative therapy can be a successful treatment modality to improve metastasis-related pain and symptoms of catecholamine excess, although the sample size is limited. Of the 8 painful lesions that had technically successful ablations and available follow-up, ablation led to improvement in metastasis-related pain in all cases. Of the 5 technically successful procedural sessions with available follow-up that were performed to treat manifestations of catecholamine excess, improvement in symptoms of catecholamine excess occurred in four (80%). Given the limited number of patients with available follow-up data who had ablations performed to treat manifestations of catecholamine excess, we did not analyze biochemical response following ablation. Excluding the prior case series from our institution, other published studies did not report on symptom outcome for patients with metastatic PPGL who were treated with ablative therapy [8,9,10,11].

With regard to procedure-related complications, we found that for the majority of patients, treatment of metastatic PPGL with ablative therapy was safe. No complications were noted following 33 (67%) procedural sessions with available follow-up, and 7 (14%) procedural sessions had only Clavien–Dindo Grade 1 complications. One patient did have a Grade IV complication; however the gastrointestinal bleeding was noted at multiple sites so it is unclear if this complication was directly caused by the RFA. Additionally, one patient died due to a rare complication of a suspected argon gas embolism during CRYO of 4 osseous lesions.

Unsurprisingly, cardiovascular monitoring for labile post-ablation hemodynamics was required following 6 (12%) procedural sessions. However, there were no long-term sequelae for these patients. Even with pre-ablation adrenergic blockade and metyrosine, significant release of catecholamines can occur during ablation. Therefore, it is essential for patients with functioning metastases to be treated carefully with pre-ablation adrenergic blockade and/or metyrosine and to have appropriate anesthesia care intra- and post-ablation.

Our study has several strengths and limitations. Some of the patients in our study were referred only for ablation and had post-ablation follow-up elsewhere, and therefore unknown long-term outcomes. However, the retrospective study design allowed us to have a relatively large sample size of patients, given the rarity of metastatic PPGL. The retrospective study design also allowed us to follow patients for many years post-ablation, which was particularly valuable for determining long-term radiographic response to treatment. A potential bias in our cohort was the treatment of patients within a single tertiary-care setting. In general, there is a number of local and systemic treatment combinations available to treat metastatic PPGL and selection of therapy is often based on the experience of the individual clinician and the practice at a specific institution. Ultimately, our cohort of patients was heterogeneous in regard to the aggressiveness and extent of disease, functional classification, and treatment with therapies in addition to ablation. The heterogeneity of our cohort reflects the population of patients with metastatic PPGL and the radiographic, symptom, and safety outcomes of our study are generalizable to patients with metastatic PPGL.

## 4. Materials and Methods

### 4.1. Patient Demographics and Clinical Presentation

To study the efficacy and safety of RFA, CRYO, and PEI in the treatment of patients with metastatic PPGL, we retrospectively reviewed medical records of a consecutive cohort of patients with metastatic PPGL. We only included patients who provided authorization to use their health records for research purposes. These patients were evaluated in the Mayo Clinic System, USA, between June 14, 1999 and November 14, 2017. The study was approved by the Institutional Review Board of Mayo Clinic, Rochester, MN (the IRB number for this project: 13-004137).

### 4.2. Subjects

The Mayo Clinic PPGL database was reviewed to identify patients with metastatic PPGL who were treated with RFA, CRYO, or PEI. Of the 273 patients identified with metastatic disease, 31 (11%) were treated with at least one thermal or chemical ablation. All of these patients treated with ablation were 17 years or older at the time of ablation. The medical records were reviewed to assess each patient’s clinical presentation, biochemical data, imaging results, procedural reports, pathology, and response to ablation. Of note, a Mayo Clinic case series published in 2011 examined 10 patients who had 47 metastatic lesions ablated with RFA, CRYO, or PEI [8]. Our study included patients from this original Mayo Clinic case series [8].

### 4.3. Disease-Related Definitions

For the purposes of this study, metastatic disease was defined according to the 2017 World Health Organization criteria [5]. A patient was considered to have synchronous metastatic disease if metastases were diagnosed within three months of the primary tumor’s discovery. Metachronous metastatic disease was defined as the development of metastases at least 3 months after the primary tumor was diagnosed. PPGL tumors were defined as functional when plasma or urine total or fractioned catecholamines or metanephrines were above the upper limit of normal for each value’s reference range. Tumors producing excess epinephrine or metanephrine were defined as adrenergic. Tumors producing excess norepinephrine or normetanephrine were defined as noradrenergic. Tumors with excess dopamine production were defined as dopaminergic. Biochemical evaluation was only included in data analysis if it was completed within three months prior to each ablation session.

### 4.4. Thermal and Chemical Ablation Overview

Our ablation procedural technique and protocol have been previously described [14,15,16]. All procedures were performed with patients under general anesthesia. Hepatic ablations at Mayo Clinic were performed with RFA or PEI. PEI was only utilized for hepatic lesions when a lesion was not amenable to thermal treatment due to close proximity to central bile ducts (<1 cm). All extra-osseous RFA treatments were performed with an impedance-based internally cooled RFA device (Cool-tip^TM^, Medtronic/Covidien, Dublin, Ireland). The STAR tumor ablation system (DFINE, San Jose, CA, USA) was used for all RFAs in the spine and bone. All CRYO treatments were performed using the Precise Cryoablation System (Galil Medical, Yokneam, Israel) or Endocare Cryotherapy System (Healthtronics, Irvine, CA, USA). The decision to use CRYO or RFA for osseous lesions was based on operator preference and consideration of patient safety.

Protective maneuvers to mitigate the risk of injury to adjacent structures were employed based on operator preference and tumor location. Hydrodissection using normal saline or sterile water was used when the bowel or skin necessitated movement for complete ablation. Computed tomography (CT) myelography was performed during spinal ablation when precise visualization of the spinal cord and adjacent nerve roots were needed. Neurophysiologic monitoring via motor or somatosensory evoked potentials by the Department of Neurology was utilized on a case-by-case basis when a tumor was closely associated with the spine or major peripheral nerves. CT and/or ultrasound guidance was utilized for probe placement: the number of probes and duration of treatment was based on tumor size, tumor location, and operator preference. CT utilizing a Siemens Somatom Sensation open 40-slice system (Siemens AG, Munich, Germany) was used for peri-procedural monitoring of treatment.

### 4.5. Radiographic Outcomes Definitions

For the purpose of this study, RFA, CRYO, and PEI were considered technically successful if the ablation defect encompassed the index tumor with no intra- or peri-tumoral enhancement on imaging performed within three months after the ablation, typically CT or MRI. Additionally, for lesions treated with CRYO, the ablation was considered technically successful if the ice ball completely encompassed the metastatic lesion during the ablation.

Post-ablation imaging was compared to pre-ablation imaging to determine each tumor’s radiographic response to therapy. Radiographic local control was assessed using the most recent post-ablation follow-up imaging study. Radiographic local control was defined as no evidence of disease in the area of ablation or decreased tumor burden in the area of ablation compared to pre-ablation imaging. A lesion was considered to have radiographic progression if tumor burden in the area of ablation increased or was stable compared to pre-ablation imaging. The same radiologist assessed the imaging studies to determine if there was local control or progression.

### 4.6. Symptom Outcomes

The two symptom outcomes studied were (1) post-ablation improvement in manifestations of catecholamine excess and (2) metastasis-related pain. A patient was considered to have manifestations of catecholamine excess if a provider attributed any number of symptoms (hypertension, headaches, anxiety, palpitations, paroxysmal spells, etc.) to the patient’s metastatic PPGL. Clinical notes were reviewed following ablation to determine if manifestations of catecholamine excess and metastasis-related pain improved following treatment. Symptom improvement was considered a categorical variable (improved/not improved).

### 4.7. Procedural Complications

Procedural complications were graded using the revised Clavien–Dindo classification system. This system defines a complication as any deviation from the normal postoperative course [17]. A Grade I complication does not require intervention outside of basic therapeutic regimens such as analgesics or anti-emetics [17]. A Grade II complication requires pharmacologic treatment with medications other than those allowed for a Grade I complication [17]. A Grade III complication requires a procedural intervention [17]. A Grade IV complication is a life-threatening event, and a Grade V complication is death [17].

### 4.8. Statistical Analysis

The data were summarized using descriptive statistics. Continuous data were presented as median and minimum–maximum range. Categorical data were presented as absolute and relative frequencies (percentages). For categorical variables, the reported frequencies (percentages) only included known outcomes unless otherwise stated. Associations between categorical variables were assessed using the chi-square test. *p* values less than 0.05 were considered significant. Data were analyzed using JMP software, version 10 (SAS, Cary, NC, USA).

## 5. Conclusions

For patients with metastatic PPGL, ablation therapy with RFA, CRYO, or PEI should be considered in the following circumstances: (1) to palliate painful abdominal/pelvic or osseous metastases when there are a limited number of culprit lesions; (2) to reduce symptoms of catecholamine excess secondary to functioning abdominal/pelvic or osseous metastases when the bulk of the disease burden can be targeted with ablation; and, (3) to achieve radiographic local control and halt progression of abdominal/pelvic or osseous metastases that are likely to cause morbidity with continued growth. Due to the rarity of metastatic PPGL and the multi-disciplinary approach required to treat patients with this disease, the patient’s best interest is served by having ablative procedures performed in high volume centers. Given the potential for serious procedure-related complications, shared decision making between clinicians and patients regarding the risks and benefits of ablative therapy is essential.

## Figures and Tables

**Figure 1 cancers-11-00195-f001:**
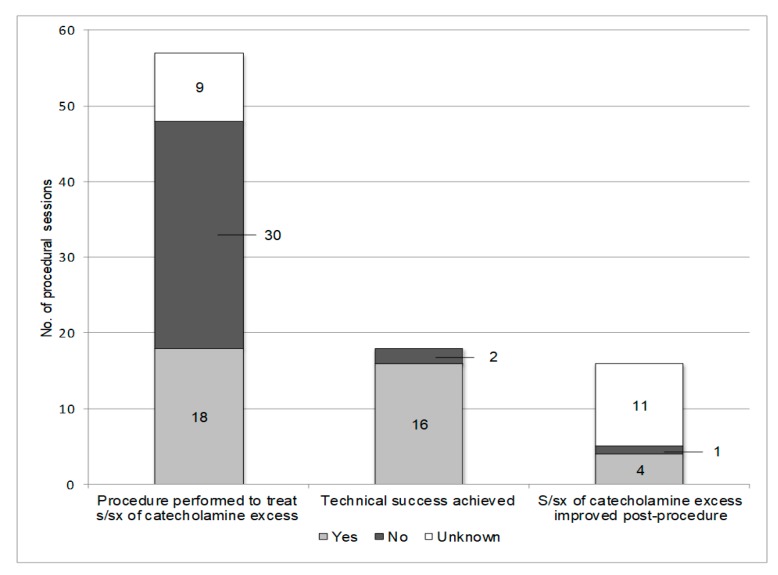
Outcomes of ablations performed to treat manifestations of catecholamine excess in patients with metastatic pheochromocytoma and paraganglioma. Eighteen procedures were performed to treat patients with manifestations of catecholamine excess. Technical success was achieved in 16 of those procedures, after which five patients had known symptom outcomes. Of those five, four patients had improvement in symptoms of catecholamine excess following ablation. Abbreviations: s/sx, signs and symptoms.

**Figure 2 cancers-11-00195-f002:**
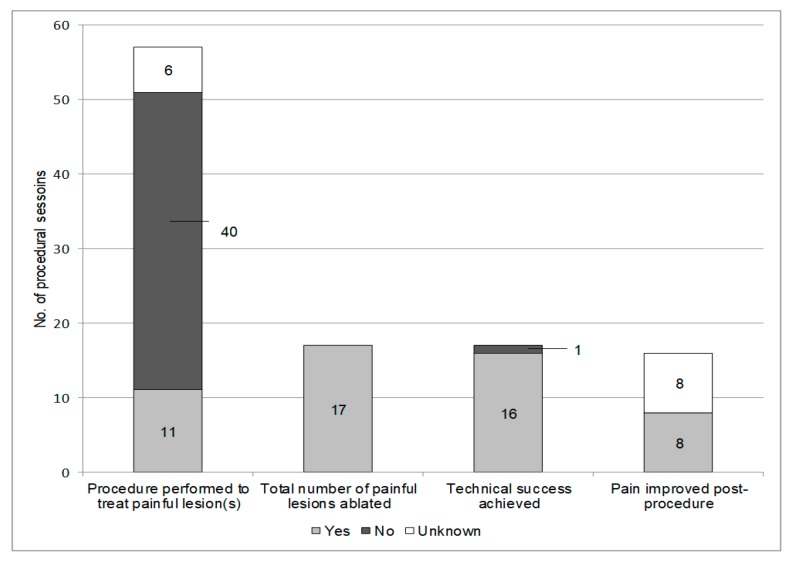
Outcomes of ablations performed to treat pain in patients with metastatic pheochromocytoma and paraganglioma. Eleven procedures were performed to treat a total of 17 painful metastases. Technical success was achieved for 16 of the ablated metastases. Of those lesions, all eight patients with symptom follow-up post procedure had improvement in pain.

**Figure 3 cancers-11-00195-f003:**
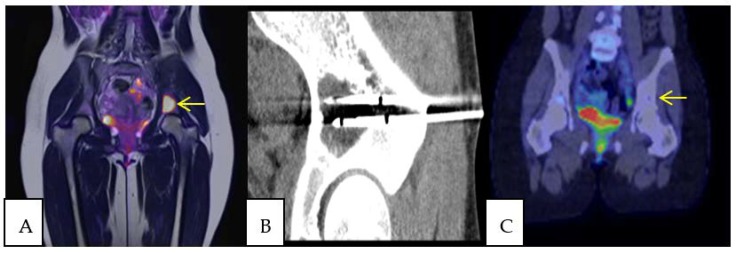
A 17 year old female with widely metastatic paraganglioma underwent CRYO and radiofrequency ablation (RFA) of multiple osseous lesions to palliate symptoms of catecholamine excess and achieve local control. Pre-ablation PET/MRI demonstrated a fluorodeoxyglucose (FDG) avid lesion involving the supraacetabular left ilium (**A**). Two cryoablation needles were placed into the lesion and the iceball encompassed the metastasis (**B**). At 20 months post procedure, PET/CT demonstrated no evidence of residual hypermetabolic paraganglioma in the supraacetabular left ilium (**C**). Additionally, following ablation her symptoms of catecholamine excess improved and she reduced the dose of her chronic adrenergic blockade.

**Table 1 cancers-11-00195-t001:** Baseline clinical characteristics of patients with metastatic pheochromocytoma and paraganglioma treated with ablative therapy. Categorical data presented as absolute and relative frequencies (percentages). Continuous data presented as median (minimum–maximum range). * B symptoms include fevers, chills, night sweats, weight loss, and anorexia. Abbreviations: mm, millimeter; *NF1*, neurofibromatosis type 1; PGL, paraganglioma; PHEO, pheochromocytoma; *SDHB*, succinate dehydrogenase subunit B; and *SDHD*, succinate dehydrogenase subunit D.

Characteristics	Data, Number (Percent or Range)
**N**	31
**Female sex**, *n* (%)	22 (71%)
**Primary tumor**, *n* (%)	
PGL PHEO	24 (77%) 7 (23%)
**Genetic status**, *n* (%)	
*SDHB* positive *SDHD* positive *NF1* positive Sporadic No genetic testing performed	17 (74%) 1 (4%) 1 (4%) 4 (17%) 8
**Age at primary tumor diagnosis**, years (range)	27 (8–72)
**Mode of primary tumor discovery**, *n* (%) Symptoms of catecholamine excess Symptoms of tumor-related mass effect Incidental discovery on imaging Symptoms of catecholamine excess + tumor mass effect Hypervascular right tonsillar mass Syncope with neck rotation B symptoms * Unknown	12 (39%) 9 (29%) 5 (16%) 2 (6%) 1 (3%) 1 (3%) 1 (3%) 1
**Primary tumor location**, *n* (%) Abdomen/pelvis Skull base and neck Thorax	31 (76%) 7 (17%) 3 (7%)
**Primary tumor hormonal status**, *n* (%) Functional Non-functional Unknown	19 (83%) 4 (17%) 8
**Primary tumor size**, mm (range)	55.5 (10–190)
**Surgical resection of the primary tumor**, *n* (%)	32 (94%)
**Age at diagnosis of metastatic disease**, years (range)	38 (12–77)
**Time to diagnosis of metastatic disease**, years (range)	4 (0–53)
**Metachronous metastases**, *n* (%)	23 (74%)
**Location of metastases**, *n* (%) Osseous Abdomen Thorax Pelvis Neck Brain	27 (90%) 23 (74%) 13 (42%) 10 (33%) 3 (10%) 1 (3%)
**Metastases per patient**, *n* (range)	>5 (1–>5)
**Treatment with systemic therapy**, *n* (%)	14 (45%)

**Table 2 cancers-11-00195-t002:** Therapeutic approaches and outcomes in patients with metastatic pheochromocytoma and paraganglioma. Categorical data presented as absolute and relative frequencies (percentages). Continuous data presented as median (minimum–maximum range). Abbreviations: mm, millimeter.

Variable	Data
**Total lesions ablated**, *n*	123
**Total procedural sessions**, *n*	57
**Ablation sessions per patient**, *n* (range)	1 (1–8)
**Total ablation sessions**, *n* (%) Radiofrequency ablation Cryoablation Percutaneous ethanol injection	69 42 (61%) 23 (33%) 4 (6%)
**Location of ablated lesions**, *n* (%) Osseous Hepatic Abdominal/pelvic, non-hepatic	63 (51%) 54 (44%) 6 (5%)
**Lesions ablated per patient**, *n*	3 (1–15)
**Size of ablated lesions**, mm (range) Osseous Hepatic Abdominal/pelvic, non-hepatic	15.5 (3–65) 15 (4–47) 21.5 (16–36)
**Metastases treated at the time of ablation**, *n* (%) Not all present metastases treated All present metastases treated Unknown	35 (76%) 11 (24%) 11
**Technical success for each ablated lesion**, *n* (%) Achieved Not achieved Unknown	94 (94%) 6 (6%) 19

**Table 3 cancers-11-00195-t003:** Radiographic outcomes of ablated metastases in patients with metastatic pheochromocytoma and paraganglioma. Categorical data presented as absolute and relative frequencies (percentages). Continuous data presented as median (minimum–maximum range). ***** One patient underwent cryoablation (CRYO) and percutaneous ethanol injection (PEI) of a single osseous lesion (right humerus) during the same procedural session. Since this lesion was treated with two ablative modalities during the same procedural session, it was excluded from analysis.

All Ablated Metastases			
Variable			Data
**Local control**, *n* (%) Radiofrequency ablation Cryoablation Percutaneous ethanol injection			69/80 (86%) 44/51 (86%) 24/28 (86%) 1/2 (50%)
**Local control**, *n* (%) Patients treated with systemic therapy Patients not treated with systemic therapy			69/80 (86%) 36/46 (78%) 33/34 (97%)
**Duration of follow-up for ablated lesions with local control**, months (range)			26 (2–163)
**Progression for ablated lesions**, *n* (%) Radiofrequency ablation Cryoablation Percutaneous ethanol injection			11/80 (14%) 7/51 (14%) 4/28 (14%) 1/2 (50%)
**Time to progression**, months (range)			16 (6–69)
**Osseous and hepatic metastases treated with radiofrequency ablation and cryoablation**			
	**Osseous**	**Hepatic**	***p* value**
N	45	32	
**Local control**, *n* (%) Radiofrequency ablation Cryoablation	37/45 (82%) 14/19 (74%) 23/26 (88%)*	30/32 (94%) 30/32 (94%) 0/0 (0%)	0.14 0.04
**Progression**, *n* (%) Radiofrequency ablation Cryoablation	8/45 (18%) 5/19 (26%) 3/26 (12%)*	2/32 (6%) 2/32 (6%) 0/0 (0%)	0.14 0.04

**Table 4 cancers-11-00195-t004:** Procedure-related complications and long-term mortality data of patients with metastatic pheochromocytoma and paraganglioma treated with ablative therapy. Procedural session complications were graded according to the revised Clavien–Dindo classification system. Categorical data presented as absolute and relative frequencies (percentages). Continuous data presented as median (minimum–maximum range). Abbreviations: PPGL, pheochromocytoma and paraganglioma.

Complication and Mortality Rates	Data, Number (Percent or Range)
**Procedural session complication rate**, *n* (%) No complication Grade I Grade II Grade III Grade IV Grade V	33 (67%) 7 (14%) 7 (14%) 0 1 (2%) 1 (2%)
**Long-term patient outcomes**	
**Deceased secondary to metastatic PPGL**, *n* (%)	10 (32%)
**Time from ablation session to death**, months (range) Session 1 (*n* = 10) Session 2 (*n* = 4) Session 3 (*n* = 3) Session 4 (*n* = 2) Session 5 (*n* = 2) Session 6 (*n* = 1)	63.5 (2–133) 68.5 (8–113) 52 (51–89) 21 (0–42) 17.5 (0–35) 26
**Alive at time of last follow-up**, *n* (%) Time from ablation session 1 to most recent follow-up, months (range)	21 (68%) 60 (0–163)

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
