# Peer review of "Efficacy and Safety of Ablative Therapy in the Treatment of Patients with Metastatic Pheochromocytoma and Paraganglioma"

_cancers, 2019, doi:10.3390/cancers11020195_

Reviewer 1 Report

Article Review: Cancers-435064

Efficacy and Safety of Ablative Therapy in the Treatment of Patients with Metastatic Pheochromocytoma and Paraganglioma

This a retrospective study of metastatic PPGL patients treated by local ablative therapy in a tertiary referral center. For this very rare disease, it is a large series of patients  which reports both efficacy and safety of local ablative procedure. The population of patient is heterogeneous, which reflects the heterogeneity of the disease. However, some additional data on whether the patients had progressive or stable disease at the moment of the treatment would be helpful.

The concept of local treatment for the control of secretion-induced symptoms is  very interesting but, part of the follow-up data are lacking to better estimate the efficacy of this strategy.

The authors are experienced in managing metastatic PPGL patients and report few severe cardiovascular: side-effects. Therefore, it would be interesting for the lectorship to have the author’s opinion/protocol on the peri-operative management of these patients which is key for better safety.

Some pictures for the 3 indications of ablative treatment listed by the authors  would be helpful, if possible: palliation of pain, local control and secretion control; this could help lectorship to picture which patient to select in their own practice.

Following are my comments/questions:

Introduction section

Page 2 line 51 : 1 of the goal of treatment is also to prolong overall survival/ enhance prognosis although I acknowledge that no study has demonstrated an increased overall survival with any of the treatment currently in use.

Results section

Table 1: percentage of unknown genetic status and unknown mode of primary tumor discovery are lacking.

Page 3 line 74-81: seems redundant with table 1, especially the data on primary tumors which does not represent the status of the treated patients so well. I would argue for summarizing section 2.2 & 2.3 in a section with the characteristics of the patients at the time they were treated so that the reader can picture what type of patients were treated.

Do the authors know if the treated patients had progressive disease at the time of treatment? Or were they stable?

Do the authors have information about scintigraphic statius of the patients? Were they FDG pET positive as most of them were SDHx mutated? Was FDG PET used for the follow-up?

Section 2.5: the authors state that ablation was performed to achieve local control and mitigate the risks of tumor progression; within this indication, were the ablation session performed in patient with progressive or stable disease.

Figure 1: the outcome of procedures performed for catecholamine excess symptoms is very interesting.

Is it really impossible to retrieve the 11 missing data for the post procedure assessment?

It would be interesting to know if these procedures were done in the eleven patients for whom all present metastases were treated (table 2) and/or of they were done as part of a multimodal treatment to decrease tumor burden or as salvage therapy in secretion refractory patients despite alfa blocker, beta blocker and metyrosine TTT?

Would it be possible to have some imaging data in order to picture the patients who had symptomatic improvement following procedure for catecholamine excess? (i.e. low or hgh tumor burden?)

Did the complications occurred in the patients treated with an anti- secretory aim?

Section 2.6: Most of the patient were treated with “secretion blockade” which is probably the reason why the rate of cardiovascular complication is low although some authors in the literature argue for less or no premedication. I think this a very important part of this study as safety is key in these patients.

Maybe this part of the safety management could be more emphasized in the discussion section and maybe the authors could provide their anti secretory standard protocol in the methods section (which molecule? Dose? For all patients? Only metaneprins positive? Etc); this could be of help for daily practice of the readers.

Section 2.8 and methods 4.5: I find the technical success definition not very well defined: the authors do not precisely state how the patient were assessed before and after the RFA, CRYO and PEI procedure. Was it by CT-scan only or have I misunderstood? Did the patient were followed-up with MRI which is probably the most sensitive technique for both liver and bone imaging and for detecting post-procedure relapse?

Was FDG PET performed? With a high number of SDHx patients this could be very informative for assessing the technical success, especially for bone non RECIST evaluable lesions. Could the authors comment this point in the discussion or add data if available in the results section?

Section 2.10: for the osseous metastases treated, it would be interested to know if Skeletal related events (SRE) were observed during the follow-up of patients who have received locat ablation technique.

Section 2.12: is median PFS from ablation to progression of any metastatic lesion available? This could help to picture whether the patients had good, intermediate or bad prognosis. Same question for overall survival rate at 1, 2 and 5 years for example.

Do the authors have data to evaluate if performing local ablative therapy can delay the intiation of systemic treatment?

DISCUSSION SECTION

11. Line 217-220, The recent study by Gravel et al on ablative therapy for bone metastasis could be discussed at this point because it is another “large” series of treated patients.

12. Line 227-236: Do the authors have any thoughts on the added value of FDG pet, especially in SDHx patients for the follow-up and for defining local control?

13. Local ablative therapy as a mean to delay systemic treatment?

CONCLUSION

14. Some illustrative examples of the 3 indications of local ablative therapy in PPGL patients would be helpful for the reader, if figures can be added and pictures are available.

Author Response

Thank you for your constructive feedback. I uploaded a Word document with a point-by-point response to the reviewer's comments.

Reviewer 2 Report

Dear Editor,

This is a well-written retrospective study which concern the efficacy and safety of ablative therapy in the treatment of patients with metastatic pheochromocytomas and paragangliomas. However, there are several methodological problems which cannot be misregarded in view to the results and especially the fact that some patients were receiving concomitantly systemic therapy and others probably had received multiples therapeutic modalities before local therapies. 

Major comments

1.    In Table 1:

- Authors should add more data (there are listed in lines 137-142 and in the discussion part line 211-214) about prior treatments especially those acting long term (for ie.PRRTs) and their potentional synergistic effect as far as the efficacy of ablation therapy. 

-       As far as the patients with unknown genetic status (n=8) did they have any genetic panel analysis which was negative or had no genetic analysis at all.

2.    In the methods part: 

-Line 296: Metastases are defined when diagnosed within 3 months of the primary tumors discovery. However what about patients who presented metastases after the 3 months of the initial metastases. ?

-Lines 325-336: The authors should clarify if radiological progression was assessed by the same radiologist and if RECIST criteria were used.

3. In the discussion part the authors could analyse further the profile of patients whose metastases presented better response to local treatments. (for ie. initial Ki-67, number of metastases, age etc).

Minor comments

-Table 4 : Despite the small number of patients could the authors present data about the PFS of the patients after local treatment ?

Author Response

(The authors gave the same response as above.)
